# Remaining Useful Life Prediction of Rolling Bearings Using GRU-DeepAR with Adaptive Failure Threshold

**DOI:** 10.3390/s23031144

**Published:** 2023-01-19

**Authors:** Jiahui Li, Zhihai Wang, Xiaoqin Liu, Zhengjiang Feng

**Affiliations:** 1Key Laboratory of Advanced Equipment Intelligent Manufacturing Technology of Yunnan Province, Kunming University of Science & Technology, Kunming 650500, China; 2Faculty of Mechanical & Electrical Engineering, Kunming University of Science & Technology, Kunming 650500, China

**Keywords:** rolling bearing, life prediction, deep learning, GRU, DeepAR

## Abstract

Aiming at the problem that a single neural network model has difficulty in accurately predicting trends of the remaining useful life of a rolling bearing, a method of predicting the remaining useful life of rolling bearings using a gated recurrent unit-deep autoregressive model (GRU-DeepAR) with an adaptive failure threshold was proposed. First, time domain and frequency domain features were extracted from the rolling bearing vibration signal. Second, its operation process was divided into a smooth operation stage and degradation stage according to the trend of the accumulated root mean square of maximum. Then, the failure threshold for different bearings were determined adaptively by the maximum of the smooth operation data. The degradation dataset of a rolling bearing was subsequently obtained. In the meantime, a GRU-DeepAR model was built to obtain predictions of the failure time and failure probability. Appropriate model parameters were determined after a large number of tests to assure the effectiveness and prediction accuracy. Finally, the trend of time series and failure times were predicted by inputting the degradation dataset into the GRU-DeepAR model. Experiments showed that the proposed method can effectively improve the accuracy of the remaining useful life prediction of a rolling bearing with good stability.

## 1. Introduction

Rolling bearings are crucial components of rotating machinery, and the stability of equipment operation is always impacted by their health status. Due to the external environment and internal factors, the performance of rolling bearings will show a declining trend during operation [1]. If the equipment is not repaired or replaced in a timely manner, the deterioration will eventually prevent it from carrying out its duty as intended, resulting in irreversible financial losses. Therefore, the remaining useful life (RUL) prediction of rolling bearings is a key link in the operation and maintenance of rotating machinery.

Prognostics health management (PHM) of mechanical equipment is the core technology for ensuring the dependability of major equipment [2]. Data-driven methods have risen to prominence for RUL prediction in recent years, as a result of the quick advancement of artificial intelligence technologies and the emergence of industrial big data [3]. Deep learning, a key technology of artificial intelligence, is frequently used in the prognosis of the lifespan of mechanical equipment. Scholars have currently proposed some fundamental prediction models and achieved good success. Yang et al. [4] proposed to fuse 1D-CNN and 2D-CNN models, using continuous wavelet transform technology, inputting the time-frequency information of rolling bearing vibrations into the neural network, extracting its degradation features, and using the data from the degradation stage to train the fusion network to predict the remaining useful life of rolling bearings. Wang et al. [5] proposed a rolling bearings RUL prediction model integrating CNN and LSTM, using CNN to extract useful information and input it into LSTM to construct health indicators and predict failure time. Zhang et al. [6] proposed a fully convolutional variational autoencoder network, using a fully convolutional neural network to improve the variational autoencoder, improving the generalization performance of the network, reducing the difficulty of feature learning, and using frequency domain signals as model inputs to predict RUL of rolling bearings. Yao et al. [7] introduced the attention mechanism into the Gated Recurrent Unit (GRU) and proposed a GRU model fused with the attention mechanism, which can effectively predict the trend of RUL of different types of rolling bearings over time. However, most of the existing prediction models only use a single type of neural network or a simple superposition of neural networks. Although the prediction accuracy can be improved to some extent, the training time of the model also increases, which reduces the efficiency of the remaining useful life prediction. It is because, generally, the combination method of models connects different kinds of network structures end-to-end directly. In the judgment method, simple data normalization is frequently used to convert all collected data into values between 0 and 1, and the failure state is judged by whether the value reaches 1, and the failure time of rolling bearings is not always the same as the end of data collection. That is, since the collected data after normalization may already contain failure data, it is generally essential to remove the failure portion of the entire life data before forecasting in the process of algorithm verification, and the algorithm’s efficiency is decreased by the increased preprocessing step. The majority of studies simply use certain prediction values for the results, but the rolling bearings degradation process exhibits some fluctuations. The probability of failure cannot be expressed by a single forecast value, and the evaluation findings contain large errors.

Aimed at the above issues, this research proposed a rolling bearing RUL prediction method based on GRU-DeepAR and adaptive failure threshold. The model proposed in this study is a structural fusion of GRU and an autoregressive process but not a superposition of the two functions. For the first time, this method adaptively determined the beginning prediction point and failure threshold using the accumulated root mean square of maximum. With a limited number of iterations, it can not only determine the failure time point of the rolling bearings, but also forecast the failure likelihood and its probability interval. By contrasting the predictions from different datasets with those from popular prediction methods, the efficiency of the method suggested in this work was demonstrated.

## 2. DeepAR Model

DeepAR is a time series forecasting method based on autoregressive recurrent neural network. A key component of the DeepAR model is the LSTM unit or GRU unit [8].

### 2.1. LSTM Model

Recurrent neural network (RNN) is a kind of neural network with the capacity for short-term memory [9]. Its neurons have the ability to receive data from themselves as well as from other neurons. There is a loop in the network structure. Using neurons with self-feedback, RNN can analyze sequential data of arbitrary lengths. An RNN variation known as LSTM is capable of effectively resolving the gradient explosion or disappearance issue [10]. Figure 1 depicts the structure of an LSTM unit, which consists of a forget gate, an input gate, and an output gate. The LSTM is expressed mathematically as follows:(1)$$f_{t}=\sigma \left(W_{f}\cdot \left[h_{t-1},x_{t}\right]\right)$$
(2)$$i_{t}=\sigma \left(W_{i}\cdot \left[h_{t-1},x_{t}\right]\right)$$
(3)$$c_{t}=\tan h\left(W_{c}\cdot \left[h_{t-1},x_{t}\right]\right)$$
(4)$$o_{t}=\sigma \left(W_{o}\cdot \left[h_{t-1},x_{t}\right]\right)$$
(5)$$C_{t}=f_{t}*C_{t-1}+i_{t}*c_{t}$$
(6)$$h_{t}=o_{t}*\tan h\left(C_{t}\right)$$
where $h_{t}$ is the output, $C_{t}$ is the internal state, $c_{t}$ is the candidate state obtained by the nonlinear function, $i_{t}$ is the input gate, $f_{t}$ is the forget gate, and $o_{t}$ is the output gate. The function of the forget gate $f_{t}$ is to control how much information needs to be forgotten in the internal state at the previous moment $C_{t-1}$, the function of the input gate $i_{t}$ is to control how much information needs to be saved in the candidate state $C_{t}$ at the current moment, and the output gate $o_{t}$ controls how much information the internal state $C_{t}$ at the current moment needs to output to the external state $h_{t}$.

### 2.2. GRU Model

Gated recurrent unit (GRU) is a variant of LSTM that balances input and forgotten information by introducing a gating mechanism [11]. The structure of GRU is more concise than that of LSTM, whose extra memory unit is removed, and the second-order nonlinear transformation is canceled at the output. The GRU is expressed mathematically as follows:(7)$$r_{t}=\sigma \left(W_{r}\cdot \left[h_{t-1},x_{t}\right]\right)$$
(8)$$z_{t}=\sigma \left(W_{z}\cdot \left[h_{t-1},x_{t}\right]\right)$$
(9)$${\tilde{h}}_{t}=\tan h\left(W_{\tilde{h}}\cdot \left[r_{t}*h_{t-1},x_{t}\right]\right)$$
(10)$$h_{t}=\left(1-z_{t}\right)*h_{t-1}+z_{t}*{\tilde{h}}_{t}$$
where $h_{t}$ is the current state, ${\tilde{h}}_{t}$ is the candidate state at the current moment, and $z_{t}$ is the update gate. Figure 2 depicts the structure of an GRU unit. The function of the update gate $z_{t}$ is to control how much information the current state needs to retain from the historical state and how much new information to accept from the candidate state. The function of the reset gate $r_{t}$ is to determine how much historical information needs to be forgotten.

Compared with the LSTM unit, the GRU unit has only two gates. Due to the reduction of parameters, the calculation amount is also reduced accordingly, which can save the training time of the model and improve the efficiency of prediction [12]. Therefore, the RUL prediction model of rolling bearings proposed in this paper is the DeepAR model based on the GRU unit.

### 2.3. GRU-DeepAR Model

The GRU-DeepAR model can not only produce precise prediction values but can also provide the probability distribution of those values, which distinguishes it from more conventional prediction models such as CNN and RNN. The traditional RNN model is expressed mathematically as follows:(11)$$h^{\left(t\right)}=f\left(h^{\left(t-1\right)},\;x^{\left(t\right)};\theta \right)$$
where $x^{\left(t\right)}$ is the external signal, $h^{\left(t\right)}$ indicates the state of the system at the time step, and θ is the parameter of the transfer function f. The GRU-DeepAR model structure is similar to RNN, which is expressed mathematically as follows:(12)$$h_{i,t}=u\left(h_{i,t-1},z_{i,t-1},x_{i,t},\Theta \right)$$
where i represents the i−th sequence, u is a recursive function in the autoregressive neural network, $h_{i,t-1}$ is the state of the previous moment, $h_{i,t}$ is the state of the current moment, $z_{i,t-1}$ is the observed value of the previous moment, $x_{i,t}$ is the covariate of the current moment, and Θ is the parameter of the GRU-DeepAR model.

Figure 3 depicts the structure of GRU-DeepAR model, in which Figure 3a is the training stage of GRU-DeepAR model. Figure 3b is the prediction stage of GRU-DeepAR model [13]. During the training stage, the model calculates the current state $h_{i,t}$ using the observed value of the previous moment $z_{i,t-1}$, the covariates of the current moment $x_{i,t}$ and the previous moment state $h_{i,t-1}$ at each moment. Then, the likelihood parameter is calculated and the observed value at the current moment is obtained. The value before the time t is the observed value, and the value following the time t is the predicted value. As a supervised learning method, the training stage requires the immediate input of values after time t. During the prediction stage, after the training is completed, the historical data from before the time t is fed into the model to obtain the initial state $h_{i,t-1}$. Then, a random sample is taken at each time step to obtain the probability distribution ${\tilde{z}}_{i,t}\sim p(\cdot |\theta_{i,t})$ of ${\tilde{z}}_{i,t}$ using the ancestral sampling method, and the predicted value is used as the following input for the time step. p(z|θ) represents the likelihood function, the parameters of which can be affinely determined from the neural network’s output $h_{i,t}$. This procedure can be repeated to acquire the sampling value from the time t to the prediction time T and calculate the prediction value of the rolling bearings degradation time series data.

## 3. RUL Prediction of Rolling Bearings Based on GRU-DeepAR Model

DeepAR is used for the first time in the field of RUL prediction of rolling bearings in the method proposed in this research based on the GRU-DeepAR model. Since the failure thresholds are adaptively computed according to the operating conditions of different rolling bearings, this method can predict the RUL in real-time during the operation. The method is divided into two parts, including state monitoring and RUL prediction. The state monitoring part includes raw data collection, signal processing, and building dataset for model training. The RUL prediction part includes model building, model training, and error evaluation. Figure 4 depicts the specific flow of the prediction process.

First, feature extraction is performed on the vibration signal. Then, the operating stages of the rolling bearings are divided according to the extracted feature values. Next, the failure thresholds are adaptively determined for the operating conditions of different rolling bearings. Then, the dataset is constructed for training. Finally, the RUL of the rolling bearings is predicted, and errors are assessed.

The specific steps of the algorithm are described as follows:Feature extraction of vibration signal

Time-domain feature, dimensionless feature, and frequency-domain feature extraction are carried out for each sample of the rolling bearings’ full-life vibration signal. The extracted features will be used to establish the bearing degradation feature sequence according to the time sequence of signal collection. Table 1 lists the time domain features, dimensionless features, and frequency domain features of the feature extraction step.

2.Adaptive determination of failure thresholds

To collect important degradation information about the rolling bearings, their entire life cycles are separated into different operation stages. Based on the results of the division, the failure thresholds are then adaptively calculated. The traditional approaches to RUL prediction use data from the entire life cycle to make predictions, but it suffers from the disadvantage that rolling bearings’ degradation characteristics are not obvious during the smooth operation stage. This results in low prediction efficiency and inaccurate prediction results. Meanwhile, the traditional method for determining the failure threshold is to normalize all data between 0 to 1 directly and take 1 as the failure threshold. The collected data are not analyzed and processed, which leads to the fact that the failure time corresponding to the failure threshold does not match the actual failure time. In order to solve the above problems, the scheme in this paper was to first divide the operation stage into a smooth operation stage and a degradation stage, determine the failure threshold using the data from the smooth operation stage and then predict the RUL using the feature data that was extracted from the degradation stage. The method of the adaptive threshold determination is as follows.

Select the maximum value from the extracted time domain features to create a time series, calculate the root mean square of the maximum value of the vibration signal from the previous minute, and calculate the mean and standard deviation of the previous root mean square. The rolling bearing starts to show a degeneration trend when the vibration signal amplitude fluctuates significantly. In 3σ principle, the root mean square of the rolling bearing maximum value deviates from $\mu_{n}+3\sigma_{n}$ as formula. Before that, all maximum values are contained within the range of $\mu_{n}+3\sigma_{n}$, and the operation state of the rolling bearing is smooth operation. Therefore, it is concluded that this point marks the beginning of the rolling bearing’s degradation stage. The process of dividing operation stages is expressed mathematically as follows:(13)$$RMS_{n}=\sqrt{\frac{\sum_{i=1}^{n}max_{n}{}^{2}}{n}}$$
(14)$$\mu_{n}=\frac{\sum_{i=1}^{n}RMS_{n}}{n}$$
(15)$$\sigma_{n}=\sqrt{\frac{\sum_{i=1}^{n}{\left(RMS_{n}-\mu_{n}\right)}^{2}}{n}}$$
(16)$$RMS_{n}>\mu_{n}+3\sigma_{n}$$
where $max_{n}$ is the maximum value of the sample in the nth minute, $RMS_{n}$ is the accumulated root mean square of $max_{n}$ from beginning to the nth minute, $\mu_{n}$ is the average of $RMS_{n}$ from beginning to the nth minute, and $\sigma_{n}$ is the standard deviation of $RMS_{n}$ from beginning to the nth minute.

The normal operating stage is the period previous to the starting point of degradation, which is where the RUL prediction begins. The data of the degradation stage are constructed as a rolling bearing degradation dataset according to time series.

The failure threshold is determined by taking the maximum value of the time-domain features extracted from the data in the smooth operation stage, in accordance with the divided rolling bearing degradation stage. The rolling bearing enters the stage of complete failure when the amplitude of the vibration signal is larger than 10 times the maximum value, according to the results of numerous bearing fatigue tests. The failure threshold of the rolling bearing, which is used to assess the rolling bearing failure time in the subsequent prediction processes, is therefore taken to be 10 times the maximum value of each rolling bearing in the smooth operation stage.

3.Model training

According to step 2, the rolling bearing degradation time series is produced using the vibration data after the beginning of the degradation stage and before the failure point, which is divided into training set and test set in a 7:3 ratio. The training set is then input into the GRU-DeepAR model for fitting.

4.RUL prediction and error evaluation

Input the test set into the trained GRU-DeepAR model to predict the degradation trend of rolling bearings according to the failure threshold determined in step 2 and then obtain the predicted value of failure time and the corresponding failure probability interval. Finally, compare the predicted value with the real value and evaluate the prediction accuracy of the GRU-DeepAR model.

Although the datasets used in this paper are all run-to-failure data of rolling bearings, this method is still applicable when it comes to predicting the RUL of running bearings in actual engineering. The RUL prediction approach in this study can split the operation stages before the rolling bearing fails and determine the failure threshold adaptively. It only forecasts the trend of future data based on the data before to a specific point. Therefore, this method can realize real-time RUL prediction of rolling bearings, including the failure time and failure probability interval of bearings in operation. The steps for real-time prediction are as follows:Collect the vibration data of rolling bearing to the current moment and extract the features of the vibration data;Divide the operation stages and determine the starting point of degradation and adaptively determine the failure threshold;Input all the data after the starting point of degradation into the proposed model for model training;Predict the amplitude after the current moment using the trained model and the predicted failure point is obtained when the amplitude is larger than the failure threshold.

## 4. Experimental Verification

### 4.1. XJTU-SY Dataset Experimental Verification

#### 4.1.1. Test Data

The XJTU-SY rolling bearing accelerated degradation public dataset from the team of Professor Yaguo Lei from the School of Mechanical Engineering of Xi’an Jiaotong University [14] was used to verify the algorithm. The testbed was composed of an alternating current (AC) induction motor, a motor speed controller, a support shaft, two support bearings (heavy duty roller bearings), a hydraulic loading system, etc., as shown in Figure 5.

The sampling frequency is 25.6 kHz (25.6 k points are collected in each second), the sampling interval is 1 min (the interval between two sampling is 1 min), and the sampling duration is 1.28 s for signal collection (each sampling last for 1.28 s), which comprises signals for both horizontal direction and vertical direction. The test bearing model is LDK UER204 and all test bearings are named after “working condition number_bearing number”. The dataset contains the vibration signals of the whole life cycle of 15 rolling bearings under three operating conditions (five bearings in each operating condition). Table 2 lists the test conditions of the rolling bearings.

In the dataset, there are primarily two types of rolling bearing deterioration trends. For example, the degradation trend of the third bearing in operating condition 1 (bearing1_3) is slow degradation, whose amplitude steadily increases after a period of stable operation until it exceeds the failure threshold, as shown in Figure 6a. The degradation trend of the first bearing in operating condition 2 (bearing2_1) is abrupt degradation, whose whole life cycle tends to operate smoothly until the amplitude suddenly increases and fails, as shown in Figure 6b. The bearing1_3 is operating at a speed of 2100 r/min with a radial force of 12 kN, and the bearing2_1 is operating at a speed of 2500 r/min with a radial force of 11 kN. To verify the validity of the prediction model for rolling bearings in different degradation modes, this experiment verified the above two trends for bearing degradation data.

The two bearings were chosen from the three operating conditions for the subsequent tests including bearing1_3 and bearing2_1, which both cover slow degradation and abrupt degradation trends to show the effectiveness of the suggested method. The horizontal direction vibration signal was used in the tests because it contains more degradation information than vertical direction vibration signal. Figure 6a shows the time domain diagram of bearing1_3, which has 158 samples. Figure 6b shows the time domain diagram of bearing2_1, which includes 491 samples.

The vibration signal of the test bearing is initially preprocessed according to the rolling bearing RUL prediction steps in Section 3. The work covers feature extraction, degradation stage division, prediction starting point determination, and failure threshold determination.

First, take the sample as the unit, extract the root mean square feature of the rolling bearing vibration signal during this period from each sample, and generate the time series in order as the dataset for the RUL prediction of the GRU-DeepAR model. Among the 20 features extracted from the dataset, root mean square is used to represent the vibration energy of the signal, which has good stability and repeatability. After observation and analysis, root mean square has a more obvious trend of change. Thus, it is more suitable to be used as an indicator of the degradation feature of rolling bearings. Then, calculate the starting point of prediction and the adaptive failure threshold according to the methods proposed in Section 3 (Formulas (13)–(16)). The results are as follows: The prediction starting point of bearing1_3 is 57 min while the failure threshold is 2.5 g; The prediction starting point of bearing2_1 is 451 min while the failure threshold is 4 g.

#### 4.1.2. GRU-DeepAR Model Training

The two sets of data are normalized separately. The first 70% of the dataset is used as the training set, and the last 30% is used as the test set, each serving as an input to the GRU-DeepAR model for prediction. This study uses MXNet and Gluon framework under Win10 to realize the proposed prediction model. MXNet is a deep learning framework proposed by Amazon in 2016. It adopts a hybrid approach of imperative programming and symbolic programming, which has the characteristics of saving video memory, fast running speed, and high training efficiency. The Gluon interface launched in 2017 made MXNet a step forward in imperative programming, and the construction of the network structure is more flexible. At the same time, the mixed programming method also makes the Gluon interface both efficient and flexible. The programming language version used in this study is Python3.6, and all experiments are carried out on a computer with Inter Core i7-10875H (2.3 GHz) CPU, 16GB RAM, and RTX2060-6G graphics card.

The parameters of the GRU-DeepAR model are set as follows after a large number of tests to assure the effectiveness and prediction accuracy of model training: the number of iterations epoch of model training is set to 20, the batch size is set to 16, and the GRU window size is set to 20, the number of GRU layers is set to 3, and the number of GRU units per layer is 128, as shown in Table 3. It can be concluded that the prediction accuracy is the highest when epoch parameter of GRU-DeepAR model is set to 20. When the epoch is larger, the prediction results will be inaccurate due to the overfitting of the model.

#### 4.1.3. Test Results

The RUL prediction values for the two bearings are achieved after the training of GRU-DeepAR model. When the amplitude of the expected data reaches 1, the bearing will be failed since the test dataset has removed the data from the complete failure state and normalized from the deterioration starting point to the failure threshold. The prediction results include failure time points and failure probability intervals. MSE and MAE are adopted as the error evaluation indexes, whose expressions are shown in Formulas (17) and (18).
(17)$$MSE=\frac{1}{n}\overset{n}{\underset{i=1}{\sum }}{\left(Y_{i}-\hat{Y_{i}}\right)}^{2}$$
(18)$$MAE=\frac{1}{n}\overset{n}{\underset{i=1}{\sum }}\left|Y_{i}-\hat{Y_{i}}\right|$$

Figure 7a displays the bearing1_3 forecast outcome. In the figure, the blue solid line represents real data in the degradation stage, the yellow solid line represents the failure threshold, the green solid line represents the time series predicted by the model, the dark green area represents the 50% failure probability interval, and the light green area represents the 90% failure probability interval. The green solid line in Figure 7a indicates that the predicted failure time point is at 92nd minute coincides with the actual failure point, and the error is 0. The MSE is 0.0941 and the MAE is 0.1916 calculated according to Formulas (17) and (18). Figure 7b displays the bearing2_1 forecast outcome. The predicted failure time point is at the 39th minute and the error is 3 min, compared to the actual failure time point of the 36th minute. The MSE is 0.0963 and the MAE is 0.1421 calculated according to Formulas (17) and (18). The model also forecasts the failure probability intervals of the two bearings. Considering the fluctuation of the failure probability intervals, the prediction results will be closer to the real value. In the case of bearing2_1, the 90% probability interval advances the predicted failure time to the 34th minute and the 50% probability interval advances the predicted failure point to the 37th minute, reducing the prediction error to less than 2 min. Figure 7 shows that the predicted time series trend of the test set is consistent with the true value. Meanwhile, the original data essentially falls within the predicted probability range of 50 to 90%, which can minimize prediction errors.

### 4.2. Self-Made Rolling Bearing Accelerated Degradation Testbed Experimental Verification

#### 4.2.1. Test Data

The vibration data of the self-made rolling bearing accelerated degradation testbed was also used to predict the RUL in order to verify the generalization performance of the GRU-DeepAR model in addition to the XJTU-SY dataset. As shown in Figure 8, the testbed is made up of an AC motor, a horizontal accelerometer, a vertical accelerometer, a coupling, a test bearing, a transmission bearing, a hydraulic loading system, etc. The test bearing model is the SKF-7406 angular contact bearing. During the whole bearing life stage, two IMI603C01 acceleration sensors are used to collect the vertical and horizontal vibration signals. The sampling frequency is set as 25.6 kHz, the sampling interval is set as 10 min, and each sampling time is set as 1 s. The collection is terminated when the maximum amplitude of the collected vibration signal samples exceeds ten times the maximum amplitude of the sample in normal operating stage. The test bearing was run to failure at a speed of 600 r/min and an axial load of 2 MPa in order to ensure consistency in the test conditions. In this test, the horizontal vibration signal was used to predict the RUL, as shown in Figure 9.

#### 4.2.2. Test Results

To construct the time series, the feature extraction and degradation stage division of the test data are carried out according to the life prediction algorithm steps 1–2 described in Section 3. Take the first 70% of the time series as the training set and the last 30% as the test set and feed them into the GRU-DeepAR model (Section 2.3 Formulas (11) and (12)) for training and time sequence prediction. The test result is shown in Figure 10. After calculation, the failure time predicted by this method is 98% of the rolling bearing’s entire life cycle. The MSE is 0.0415 and the MAE is 0.1647 calculated according to Formulas (17) and (18) in Section 4.1.3. The experiment demonstrates that the GRU-DeepAR model’s prediction error for the data collected by the self-made testbed is low and that it can predict the RUL of rolling bearings accurately.

### 4.3. Comparative Experiment

The major RUL prediction intelligent models of rolling bearings CNN [15,16,17] and LSTM [18,19,20] were chosen to forecast the RUL of three bearings in the above two datasets to demonstrate the superiority of the prediction method in this study. Figure 11, Figure 12 and Figure 13 display the experiment results.

The RUL prediction result of the comparison model CNN for bearing1_3 in the XJTU dataset is shown in Figure 11a. The prediction error of the model is 5 min; The MSE is 0.1691 and the MAE is 0.2169 calculated according to Formulas (17) and (18) in Section 4.1.3. The RUL prediction result of the comparison model LSTM for bearing1_3 in the XJTU dataset is shown in Figure 11b. The predicted values did not reach the failure threshold, and the model did not predict the failure point; The MSE is 0.1898 and the MAE is 0.2700 calculated according to Formulas (17) and (18) in Section 4.1.3. Compared with the prediction results of the GRU-DeepAR model for the bearing1_3 of the XJTU-SY dataset in Section 4.1.3, the prediction error of the model is 0, the MSE is 0.1691 and the MAE is 0.2169, which are lower than that of other two models. Therefore, the prediction error of the GRU-DeepAR model for the XJTU-SY bearing1_3 is lower, and the accuracy is higher.

The RUL prediction result of the comparison model CNN for bearing1_3 in the XJTU dataset is shown in Figure 12a. The predicted values did not reach the failure threshold, and the model did not predict the failure point; The MSE is 0.1437 and the MAE is 0.1929 calculated according to Formulas (17) and (18) in Section 4.1.3. The RUL prediction result of the comparison model LSTM for bearing1_3 in the XJTU dataset is shown in Figure 12b. The prediction error of the model is 3 min; The MSE is 0.1698 and the MAE is 0.1559 calculated according to Formulas (17) and (18) in Section 4.1.3. Compared with the prediction results of the GRU-DeepAR model for the bearing2_1 of the XJTU-SY dataset in Section 4.1.3, the prediction error of the model is 3 min, the MSE is 0.0963 and the MAE is 0.1421. Although the failure time prediction error of the GRU-DeepAR model is the same as that of the LSTM model, the MSE and the MAE are both lower than the prediction results of the CNN model and the LSTM model. Therefore, the accuracy of the GRU-DeepAR model for the XJTU-SY bearing2_1 is higher.

The RUL prediction result of the comparison model CNN for self-made rolling bearing accelerated degradation testbed dataset is shown in Figure 13a. The predicted values did not reach the failure threshold, and the model did not predict the failure point; The MSE is 0.2637 and the MAE is 0.1812 calculated according to Formulas (17) and (18) in Section 4.1.3. The RUL prediction result of the comparison model LSTM for self-made rolling bearing accelerated degradation testbed dataset is shown in Figure 13b. The predicted values did not reach the failure threshold, and the model did not predict the failure point; The MSE is 0.3736 and the MAE is 0.2489 calculated according to Formulas (17) and (18) in Section 4.1.3. Compared with the prediction results of the GRU-DeepAR model for self-made rolling bearing accelerated degradation testbed dataset in Section 4.2.2, the prediction error of the model is 2% of the rolling bearing’s entire life cycle, the MSE is 0.0415 and the MAE is 0.1647, which are lower than that of other two models. Therefore, the prediction error of the GRU-DeepAR model for self-made rolling bearing accelerated degradation testbed dataset is lower and the accuracy is higher.

Table 4 and Figure 14 exhibit the prediction accuracy for the three rolling bearing datasets of the comparison of CNN, LSTM, and GRU-DeepAR models. It is obvious that the error evaluation indexes of the GRU-DeepAR model is lower than those of the CNN and the LSTM models. Therefore, the GRU-DeepAR model proposed in this study has higher accuracy in the RUL prediction of rolling bearings.

## 5. Analysis and Discussion

According to the experiment results of the XJTU-SY dataset and the self-made testbed dataset, the GRU-DeepAR prediction model proposed in this study can predict the degradation trend and failure time of rolling bearings more accurately, and the prediction error can be controlled in 2 min. The operating stages of rolling bearings were divided before creating the time series dataset, allowing for quick identification of the operating stages containing useful degradation information. By determining the starting point of RUL prediction, bearing data were not entered into the model when no degradation was occurring, which can improve the efficiency of lifetime predictions. In comparison to the mainstream method of determining the starting point of degradation, the method in this study can achieve automatic determination of the starting point of degradation and reduce the errors generated during the manual calibration process. Simultaneously, the failure threshold of the rolling bearing was adaptively set according to the judgment of mutation point of the MSE with the 3σ principle, which can reduce the impact of external operating conditions and its own structure on the RUL prediction, bringing the predicted failure time closer to the actual operating conditions. The prediction of the probability interval takes into account the randomness in the rolling bearing’s degradation process and extends the predicted value of the failure time to a certain range, allowing the equipment to monitor its operating status more flexibly, and repair or replace the rolling bearing before it fails completely, reducing equipment failure and economic loss. When compared to other prediction models, the GRU-DeepAR model obtained prediction results quickly and with fewer iterations, reducing the time required for RUL prediction.

## 6. Conclusions

This study proposed a RUL prediction method for rolling bearings based on a GRU-DeepAR model and adaptive failure threshold to address the problem that a single neural network model has difficulty in accurately predicting the trend of the RUL of rolling bearings. To split the degradation stages, a strategy that combined the 3σ principle and the trend of the mean square error of maximum in the time domain of the signal was suggested. In contrast to the method of setting labels in units of time, the method in this study can construct a GRU-DeepAR model based on the time series dataset of the rolling bearing degradation process, predict the time series of degradation data, estimate the rolling bearing failure time and the 50% and 90% failure probability intervals. The conclusions of this study are as follows:Effective degradation information can be rapidly extracted as model input by dividing the bearing operation stage into smooth operation and degradation stage;The results of RUL prediction of the rolling bearings under various operating conditions and degradation modes are accurate using the adaptive failure thresholds;The GRU-DeepAR prediction model has good generalization and stability in the RUL prediction of rolling bearings, which improves the prediction accuracy;In actual engineering, since the determination of the failure threshold only depends on the data in the smooth operation stage, real-time RUL prediction of rolling bearings can be realized even before the entire life data of the bearing is obtained.

In summary, the proposed RUL prediction method based on GRU-DeepAR and an adaptive failure threshold can predict the remaining useful life of the rolling bearings more accurately, and has a certain positive effect on the monitoring and maintenance of rolling bearings.

## Figures and Tables

**Figure 1 sensors-23-01144-f001:**
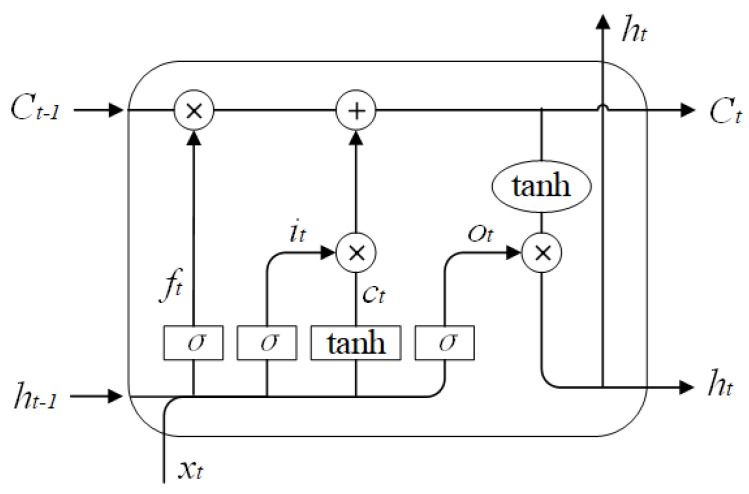
Structure diagram of LSTM.

**Figure 2 sensors-23-01144-f002:**
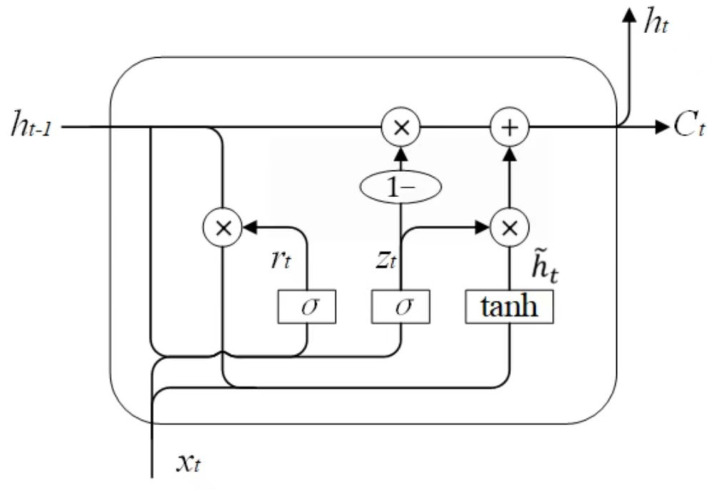
Structure diagram of GRU.

**Figure 3 sensors-23-01144-f003:**
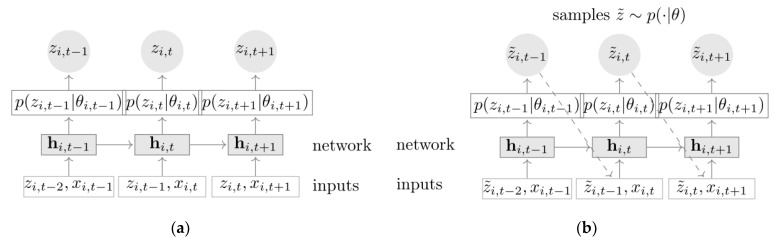
Structure diagram of GRU-DeepAR model [13]: (**a**) training stage of GRU-DeepAR model; and (**b**) prediction stage of GRU-DeepAR model.

**Figure 4 sensors-23-01144-f004:**
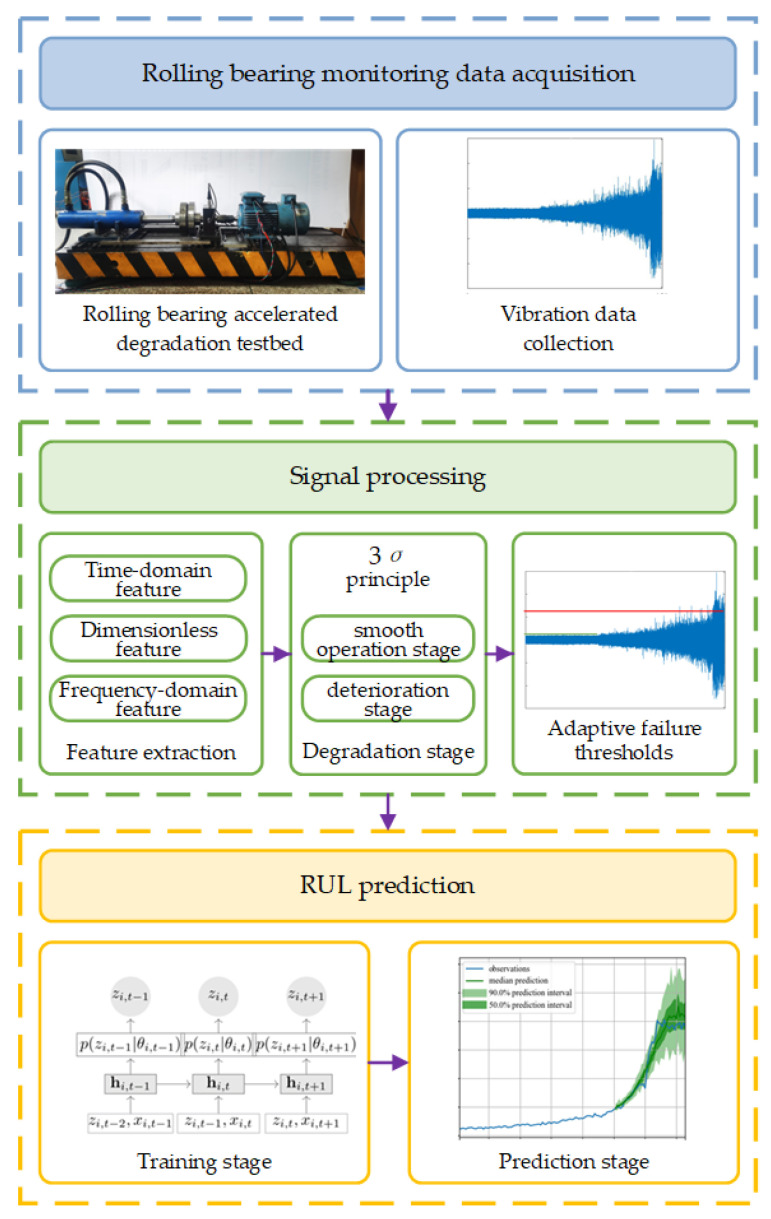
Flowchart of rolling bearings RUL prediction.

**Figure 5 sensors-23-01144-f005:**
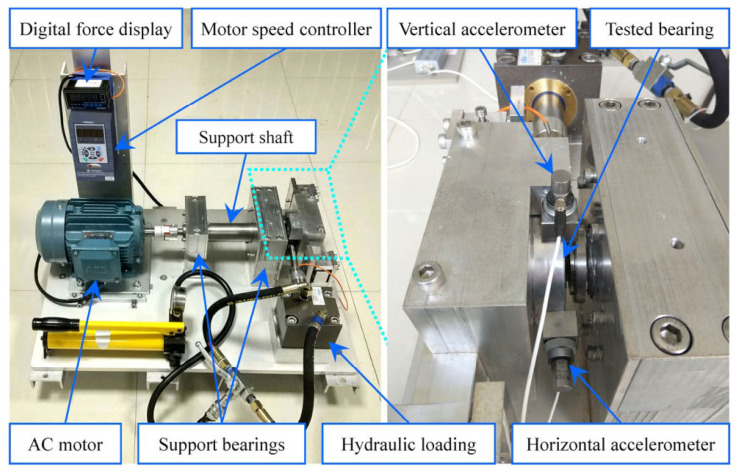
XJTU-SY rolling bearing accelerated degradation testbed [14].

**Figure 6 sensors-23-01144-f006:**
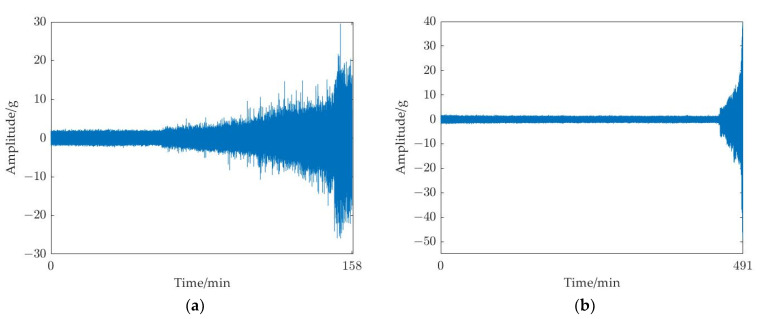
Time domain diagram of rolling bearings for test: (**a**) bearing1_3; and (**b**) bearing2_1.

**Figure 7 sensors-23-01144-f007:**
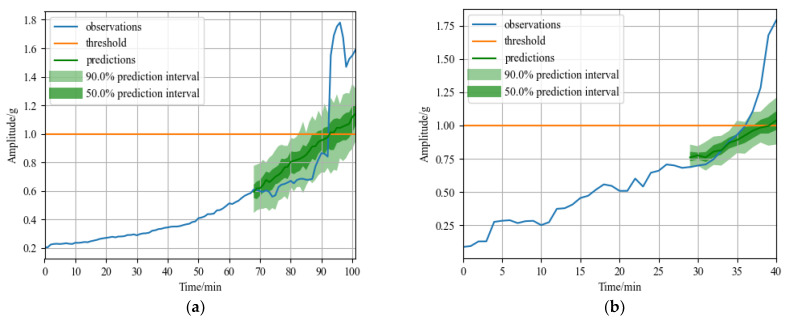
RUL prediction results of XJTU-SY dataset: (**a**) RUL prediction result of rolling bearing1_3; and (**b**) RUL prediction result of rolling bearing2_1.

**Figure 8 sensors-23-01144-f008:**
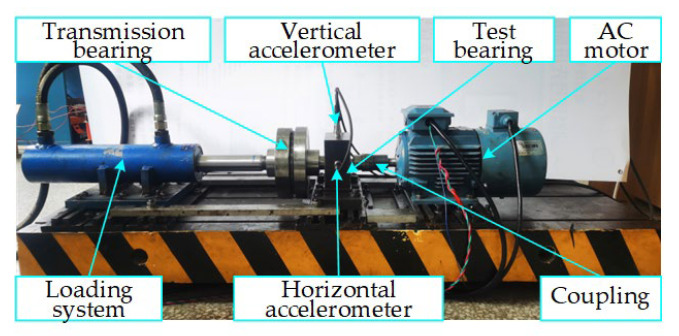
Self-made rolling bearing accelerated degradation testbed.

**Figure 9 sensors-23-01144-f009:**
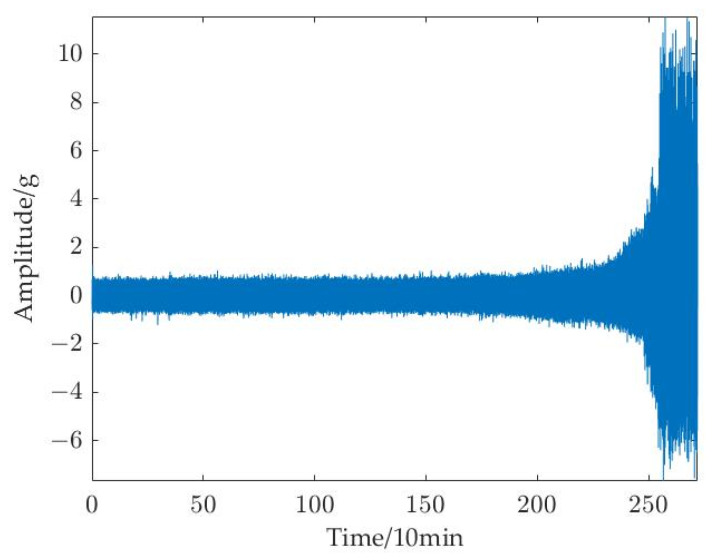
Time domain diagram of the rolling bearing on self-made testbed.

**Figure 10 sensors-23-01144-f010:**
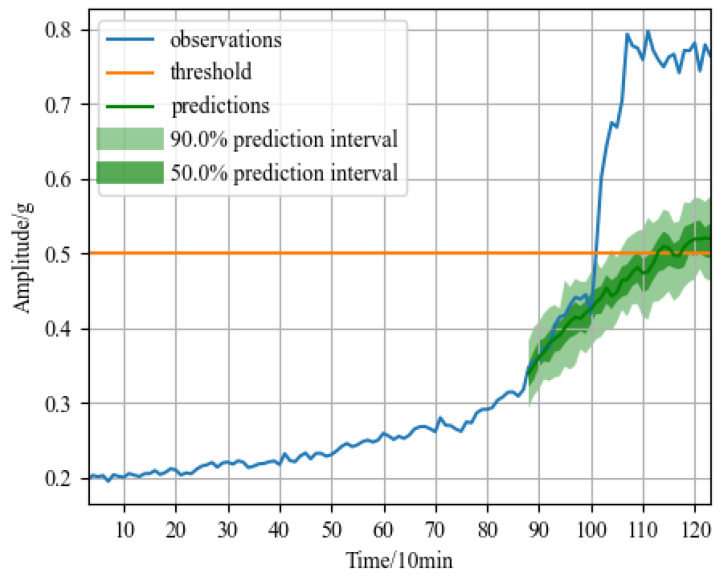
RUL prediction result of rolling bearing on self-made testbed.

**Figure 11 sensors-23-01144-f011:**
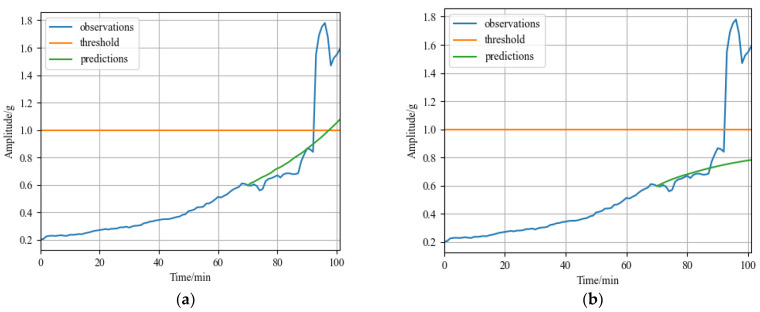
Comparative experiment results of bearing1_3 in XJTU-SY dataset: (**a**) RUL prediction result of CNN model; and (**b**) RUL prediction result of LSTM model.

**Figure 12 sensors-23-01144-f012:**
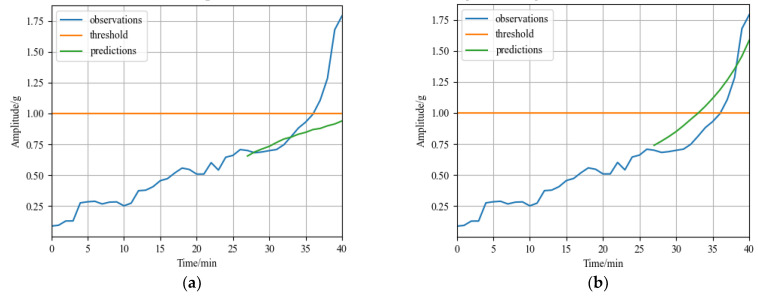
Comparative experiment results of bearing2_1 in XJTU-SY dataset: (**a**) RUL prediction result of CNN model; (**b**) RUL prediction result of LSTM model.

**Figure 13 sensors-23-01144-f013:**
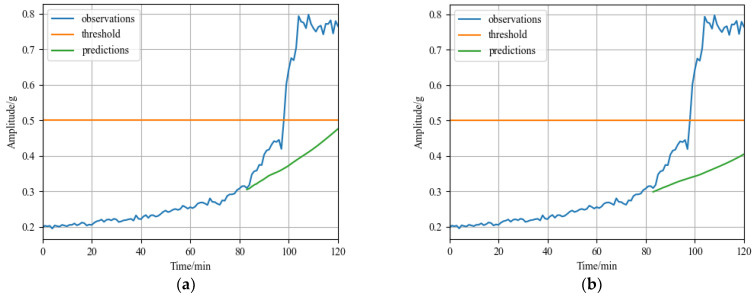
Comparative experiment results of self-made rolling bearing accelerated degradation testbed dataset: (**a**) RUL prediction result of CNN model; and (**b**) RUL prediction result of LSTM model.

**Figure 14 sensors-23-01144-f014:**
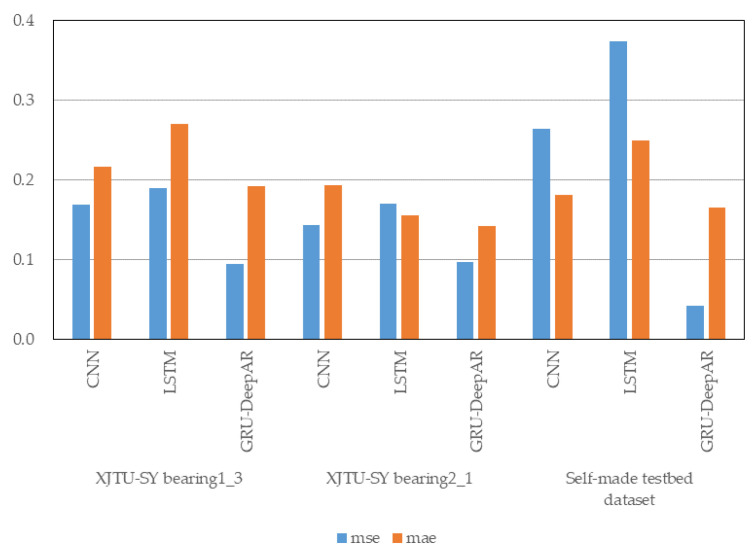
Comparison of prediction accuracy of different models.

**Table 1 sensors-23-01144-t001:** Feature extraction of rolling bearing vibration signals.

Type	Feature
time-domain feature	maximum
	minimum
	average
	peak-to-peak
	square root magnitude
	rectified average
	variance
	kurtosis
	skewness
	root mean square
dimensionless feature	waveform factor
	peak factor
	impulse factor
	clearance factor
frequency-domain feature	mean square spectrum
	root mean square spectrum
	frequency domain average amplitude
	frequency domain variance
	frequency domain peak
	center of gravity frequency

**Table 2 sensors-23-01144-t002:** Test conditions for XJTU-SY dataset rolling bearings.

Operating Condition	Rotational Speed/(r/min)	Radial Force/(kN)
condition 1	2100	12
condition 2	2500	11
condition 3	2400	10

**Table 3 sensors-23-01144-t003:** GRU-DeepAR model parameters.

Epochs	Batch_size	Window_size	Layers	Units
20	16	20	3	128

**Table 4 sensors-23-01144-t004:** Comparison of prediction accuracy of different models.

Dataset	Dataset	MSE	MAE
XJTU-SY bearing1_3	CNN	0.1691	0.2169
	LSTM	0.1898	0.2700
	GRU-DeepAR	0.0941	0.1916
XJTU-SY bearing2_1	CNN	0.1437	0.1929
	LSTM	0.1698	0.1559
	GRU-DeepAR	0.0963	0.1421
Self-made testbed dataset	CNN	0.2637	0.1812
	LSTM	0.3736	0.2489
	GRU-DeepAR	0.0415	0.1647

## Data Availability

The data presented in this study are available on request from the corresponding author.

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
