# Peer review of "Remaining Useful Life Prediction of Rolling Bearings Using GRU-DeepAR with Adaptive Failure Threshold"

_sensors, 2023, doi:10.3390/s23031144_

Round 1

Reviewer 1 Report

The manuscript ‘Sensors-2126469’ demonstrates a remaining useful life prediction method for rolling bearings with deep learning technology, which is in line with the current development direction of "industrial big data". The authors realized the RUL prediction of rolling bearings based on the combination of GRU model and DeepAR model, which have great advantages for the prediction of time series data, so the research of this manuscript is of great significance for the health management of mechanical equipment. I recommend this manuscript for publication after the authors clearly explain the following issues.

1.         Please supplement the formula description of LSTM model in section 2.1, and further improve Figure 1 to describe the structure of LSTM model in more detail.

2.         In paragraph 6 of section 4.1.1, why to choose square root magnitude as model input in?

3.         The epoch parameter of GRU-DeepAR model is set to 20 in section4.1.2. Could it be fully used for the model training stage?

4.         Please explain Formula 9, why does MSE appear sudden change when rolling bearing is in the degenerating stage?

5.         To quote pictures, tables, and formulas, hyperlinks should be used in the main text.

Author Response

Dear Reviewer,

We sincerely thank you for the time spent and the effort made in the review process of our paper. We greatly appreciate your constructive comments and suggestions on our manuscript entitled “Remaining Useful Life Prediction of Rolling Bearings using GRU-DeepAR with Adaptive Failure Threshold” (ID: sensors-2126469). All comments are very helpful for revising and improving our paper. We have studied your comments carefully and have made extensive modification which marked in red in the revised manuscript. We have tried our best to revise our manuscript according to the comments. Please see the attachment.

Reviewer 2 Report

In this paper, a method of predicting the remaining useful life of rolling bearings using Gated Recurrent Unit-Deep Autoregressive model (GRU-DeepAR) with adaptive failure threshold is proposed. The operation process is divided into smooth operation stage and degradation stage according to the trend of accumulated mean square error of maximum. Failure threshold for different bearing is determined adaptively by maximum of the smooth operation data. After that, the trend of time series and failure time are predicted by inputting the degradation dataset into GRU-DeepAR model. The proposed method is validated through experiments. The topic is interesting, however some aspects in the manuscript can be better discussed to improve its readability. The main suggestions are listed as follows:

1: In Introduction, the authors claim that although the prediction accuracy can be improved to some extent, the training time of the model also increases, which reduces the efficiency of life prediction. Is that means only using a single type of neural network or a simple superposition of neural networks will reduce the efficiency of life prediction? How the method in this manuscript deal with this problem?

2: In 2.2, the authors state that because the parameters in GRU unit is less than that in LSTM unit, the calculation amount is reduced. However, the authors haven’t supplied experimental results or references which support this statement.

3: There are spelling mistakes in this manuscript, such as “Figure 2” in 2.3.

4: In 3, the authors claim that since the failure thresholds are adaptively computed according to the operating conditions of different rolling bearings, the method in this manuscript can predict the RUL in real-time during the operation. However, how to prove that the adaptive failure threshold can make the method in this manuscript predict the RUL in real-time during the operation?

5: In the step 2 of the algorithm in 3, only the explanation of why the operation stage should be divided into a smooth operation stage and a degradation stage is provided, but there is a lack of explanation of why adaptive failure threshold should be adopted.

6: In the step 2 of the algorithm in 3, can the calculation of mean and standard deviation of previous MSE be in the form of pseudo code or equation? These may enhance the theoretical nature of this manuscript and improve the readability.

7: In the step 2 of the algorithm in 3, what are the deterioration stage and the stage of complete failure mean? What is the relationship among degradation stage, smooth operation stage mentioned previously in the manuscript and them? Can the authors illustrate the deterioration stage and the stage of complete failure in another way to avoid ambiguity if there are differences among them?

8: In the step 2 of the algorithm in 3, how can the bearings predict their own failure time and failure probability interval?

9: In the step 2 of the algorithm in 3, the meaning of “The life prediction method in this paper only predicts the changing trend of future data based on the data before a certain moment and can divide the degradation stage before the bearing fails and determine the failure threshold adaptively.” is the same as that of The RUL prediction approach in this study can split the operation stages before the rolling bearing fails and determine the failure threshold adaptively.” Why do the authors repeat twice?

10: In 4.1.1, the authors should provide the meaning of sampling frequency, sampling interval and sampling duration.

11: In Conclusions, the authors state that the adaptive failure threshold setting allows for more flexible RUL prediction of the rolling bearings under various operating conditions and degradation modes. However, this statement isn’t supported with experimental results. The authors should verify this statement with additional comparative experiments.

Author Response

Dear Reviewer,

We thank you very much for giving us an opportunity to revise our manuscript, and we greatly appreciate your positive and constructive comments and helpful suggestions on our manuscript entitled “Remaining Useful Life Prediction of Rolling Bearings using GRU-DeepAR with Adaptive Failure Threshold” (ID: sensors-2126469). Those comments are very helpful for revising and improving our paper, as well as the important guiding significance to other research. We have studied the comments carefully and made corrections which we hope meet with approval. The responds to your comments are as follows. The revised contents are marked in red in the revised manuscript. Please see the attachment.
